# DETECTING VARIANT CONTAMINATION IN LLMS VIA VARIANCE OF GENERATION DISTRIBUTION

## ABSTRACT

Evaluating large language models (LLMs) is increasingly confounded by *variant contamination*: the training corpus contains semantically equivalent yet lexically or syntactically altered versions of test items. Unlike verbatim leakage, these paraphrased or structurally transformed variants evade existing detectors based on sampling consistency or perplexity, thereby inflating benchmark scores via memorization rather than genuine reasoning. We formalize this problem and introduce **DVD** (**D**etection via **V**ariance of generation **D**istribution), a single-sample detector that models the local output distribution induced by temperature sampling. Our key insight is that contaminated items trigger alternation between a *memory-adherence* state and a *perturbation-drift* state, yielding abnormally high variance in the synthetic difficulty of low-probability tokens; uncontaminated items remain in drift with comparatively smooth variance. We construct the first benchmark for variant contamination across two domains—Omni-MATH and SuperGPQA—by generating and filtering semantically equivalent variants, and simulate contamination via fine-tuning models of different scales and architectures (Qwen2.5 and Llama3.1). Across datasets and models, **DVD** consistently outperforms perplexity-based, Min-$k$% probability, edit-distance (CDD), and embedding-similarity baselines, while exhibiting strong robustness to hyperparameters. Our results establish variance of the generation distribution as a principled and practical fingerprint for detecting variant contamination in LLM evaluation.

## 1 INTRODUCTION

In recent years, the capabilities of large language models (LLMs) have experienced explosive growth, demonstrating transformative potential across a wide range of domains (Brown et al., 2020; Team et al., 2024; Touvron et al., 2023; Chowdhery et al., 2023; Achiam et al., 2023). However, the outstanding performance of these models relies heavily on large-scale web corpora, which brings a long-standing challenge to the forefront: data contamination (Balloccu et al., 2024; Li et al., 2023; Chang et al., 2024; Cheng et al., 2025; Deng et al., 2023; Xu et al., 2024). Data contamination refers to the unintended overlap between training data and evaluation benchmarks, which severely undermines the validity of evaluation results (Cheng et al., 2025). Such overlap creates a false impression of generalization capability and may mislead research directions. When contaminated models are applied in serious scientific exploration or real-world applications, their underlying biases and flaws can lead to erroneous scientific conclusions or even catastrophic decisions, ultimately hindering technological progress (Sainz et al., 2023).

Existing detection methods—based on sampling consistency or perplexity—are useful for catching verbatim memorization but fail against a more subtle and insidious form: variant contamination. As illustrated in Figure 1, variant contamination arises when training data contains semantically equivalent but lexically or structurally altered versions of benchmark questions. Unlike exact duplicates, these paraphrased or restructured variants evade current detection approaches while still allowing models to "memorize" solutions. This phenomenon has become increasingly common as large-scale data augmentation and synthetic data generation (e.g. via GPT-4o) are widely adopted (Patel et al., 2021; Dumpala et al., 2024; Wei et al., 2025; Rabinovich et al., 2023). Consequently, high benchmark scores may reflect contamination-driven recall rather than genuine reasoning.

To investigate this challenge, we conduct a systematic study of variant contamination across multiple domains. Using Omni-MATH (mathematical reasoning) (Gao et al., 2024) and SuperGPQA (general reasoning) (Du et al., 2025), we construct a benchmark by generating semantically equivalent variants with controlled transformations. Fine-tuning models of different scales and architectures on these contaminated datasets reveals striking results: models achieve artificially high benchmark accuracy even when only variants (and not exact duplicates) are present in training. More importantly, widely used detection methods fail to flag such cases—perplexity (Li et al., 2023) and Min-K% (Shi et al., 2023) methods degrade to near-random performance (AUC < 0.5 in some settings), and distributional edit-distance methods (CDD) (Dong et al., 2024) prove unstable across domains. These findings demonstrate both the pervasiveness and the dangers of variant contamination in today's evaluation ecosystem.

To address this gap, we propose **D**etection via **V**ariance of generation **D**istribution (DVD), a novel framework that detects contamination by analyzing the fluctuations in a model's output distribution at the single-sample level. Specifically, DVD repeatedly samples answers to the same test question under controlled stochastic decoding and measures the variance of "synthetic difficulty," defined from the log-probabilities of low-likelihood tokens. The intuition is straightforward: For uncontaminated questions, the model must genuinely reason, leading to diverse but comparably uncertain outputs. This yields a relatively smooth and stable variance profile. For contaminated questions, the model alternates between recalling memorized answers (high confidence, low difficulty) and drifting into non-memorized states (higher difficulty). This mixture produces an abnormally sharp variance spike, serving as a fingerprint of contamination. Compared with existing methods, DVD goes beyond surface-level similarity (e.g., edit distance in CDD) or global likelihood measures (e.g., perplexity, Min-K%). By directly modeling distributional variance rather than absolute probabilities or token overlaps, DVD captures the hidden dynamics of memorization versus reasoning. This design makes it robust to paraphrased or structurally altered variants that elude prior approaches.

Extensive experiments show that DVD consistently outperforms baselines across datasets, domains, and model sizes. For example, on SuperGPQA, DVD improves AUC by up to +0.22 over the strongest baseline (embedding similarity), while maintaining stable performance from 1.5B to 7B parameter scales and across Qwen and Llama architectures. Sensitivity analyses further confirm that DVD is robust to hyperparameter choices, with detection performance remaining stable across wide ranges. Together, these results establish DVD as a principled and effective solution to the overlooked but critical problem of variant contamination.

Our main contributions are summarized as follows:

**Problem Identification and Formalization.** We are the first to systematically articulate the problem of variant contamination, a subtle yet widespread form of data leakage where semantically equivalent but lexically/syntactically diverse variants of benchmark items appear in training data. We provide a formal definition of this phenomenon, reveal its prevalence through benchmark construction, and demonstrate that existing detection methods fail to identify it.

**Benchmark Construction for Systematic Evaluation.** We construct the first dedicated benchmark for variant contamination detection, spanning two representative domains: Omni-MATH (mathematical reasoning) and SuperGPQA (general reasoning). Using controlled variant generation and filtering, the benchmark enables rigorous and reproducible evaluation of contamination detection methods across models, scales, and domains.

**Novel Detection Framework (DVD).** We introduce DVD (Detection via Variance of generation Distribution), a new method that leverages the variance of synthetic difficulty across multiple stochastic generations. By capturing the alternation between memory adherence and perturbation drift states, DVD provides a principled and training-data-independent indicator of contamination. Through extensive experiments, we show that DVD consistently outperforms existing baselines. Our analysis also highlights the robustness of DVD across domains and hyperparameters, ensuring stable and practical applicability in real-world evaluation pipelines.

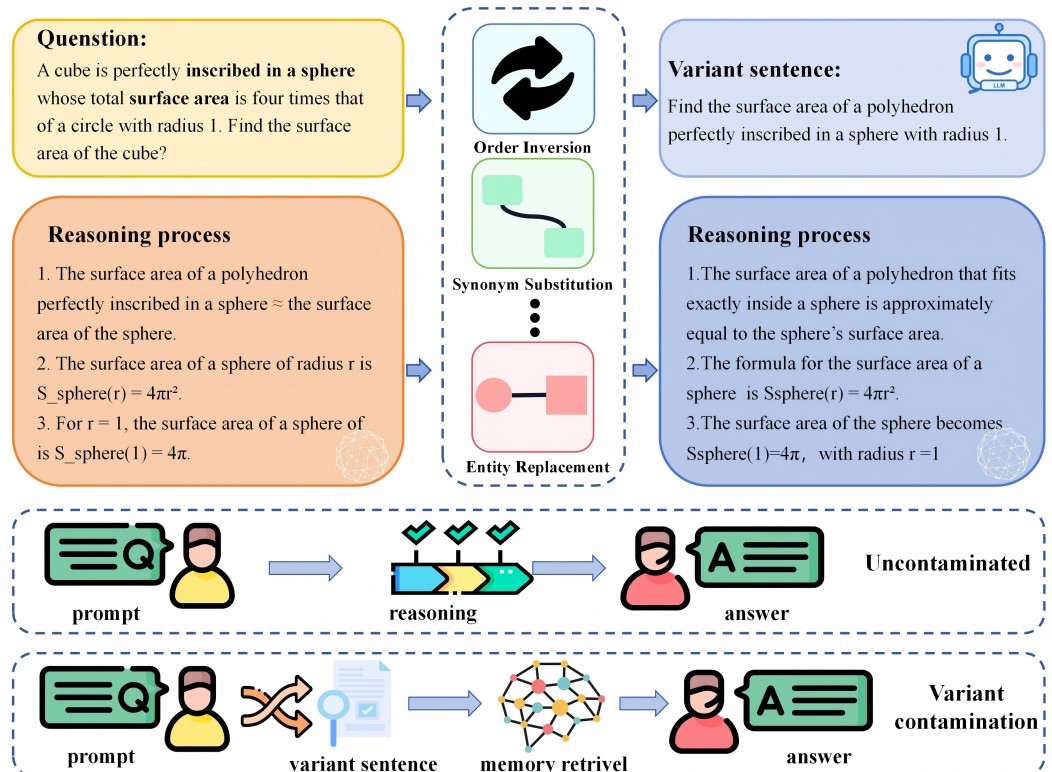

Figure 1: Conceptual illustration of variant contamination in geometric problem-solving

## 2 RELATED WORK

Existing approaches for data contamination detection can be broadly divided into two categories.

**Sampling and Output-Matching-Based Methods**   This line of research primarily relies on the similarity between model generations and reference answers, or on detecting anomalous patterns within the output distribution. Representative works include reference-instance matching based on overlap measures Golchin & Surdeanu (2023); the CDD method, which conducts multiple random samplings alongside one greedy decoding under the same prompt, and uses the edit distance between greedy and stochastic outputs to approximate the output distribution and detect sharp modes caused by memorization Khandelwal et al. (2019); and the DCQ method, which compares model preferences between original inputs and their perturbed variants to identify contamination Golchin & Surdeanu (2025). Moreover, membership inference has also been applied in this context, where the loss difference between a target sample and synthetic neighbors serves as an indicator of contamination Mattern et al. (2023). Overall, these methods are effective for detecting verbatim memorization, yet remain fundamentally limited by their reliance on shallow surface-level measures. For instance, CDD only leverages text edit distance without modeling the underlying generative probability space, thus lacking robustness against semantic variations such as paraphrasing or translation.

**Perplexity-Based Methods**   In contrast to sampling- and matching-based approaches, another class of methods focuses on detecting contamination through the abnormally high confidence that models assign to seen samples. For example, the MIN-K% PROB method examines the average log-likelihood of low-probability tokens to determine whether a sample appears in the training set Shi et al. (2023). Similarly, Oren et al. (2023) demonstrates that a model's ability to recall the order of training samples itself constitutes strong evidence of data leakage. Compared to the former category, perplexity-based methods provide a more direct quantification of model bias toward training data. However, their effectiveness is likewise constrained to verbatim memorization; once

samples undergo semantic rewriting or structural perturbation, perplexity-level differences are often obscured, leading to a significant drop in detection performance.

**Our Approach**    Motivated by the limitations of the above methods, we propose the DVD approach, which overcomes the dependence on shallow similarity measures or overall perplexity levels. Although CDD also relies on multiple samplings to construct an output distribution, its core remains restricted to edit-distance-based comparisons, failing to capture the true probabilistic dynamics underlying text generation. In contrast, DVD employs temperature sampling to generate multiple responses and systematically analyzes the variance of low-probability tokens, defined as synthetic difficulty. The key insight is that contaminated samples alternate between a "memorization-dependent state" and a "perturbation-drift state," resulting in substantially higher variance across generations. Uncontaminated samples, by contrast, remain consistently in the drift state, with variance reflecting only natural noise. By incorporating variance decomposition into a mixture-distribution framework, DVD fundamentally captures these deep probabilistic dynamics, thereby achieving superior performance in detecting semantic-variant contamination compared to existing methods.

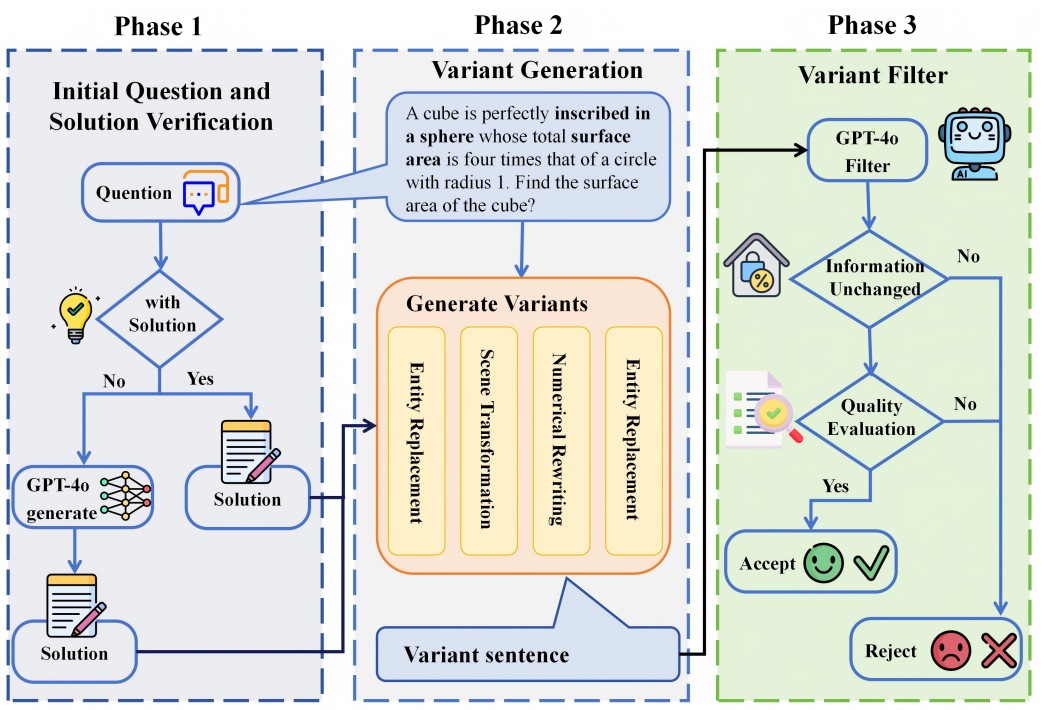

Figure 2: Semantic Equivalence Variant Generation Pipeline

## 3 VARIANT CONTAMINATION

This section introduces the formal definition of the **Variant Contamination Detection (VCD)** task (3.1) and describes the construction of a benchmark dataset tailored for variant contamination detection (3.2).

### 3.1 TASK DEFINITION

We define **variant contamination** as the scenario in which, during training, a model is exposed to samples that are logically equivalent to those in the test set but differ in surface form. Such variants may diverge in semantics, syntax, or narrative style, yet preserve the same underlying solution space, thereby allowing the model to perform as if it had previously observed the test instance.

Formally, let $x$ denote a test instance and let $f$ be a semantic abstraction function that extracts the core informational content of $x$. A variant of $x$ is then defined as:

$$v = \tau(x), \quad \text{such that } f(v) = f(x), \tag{1}$$

where $\tau$ is a transformation preserving the core semantics of $x$. If such a variant $v$ appears in the training corpus of model $M$, we say that $M$ is contaminated on test instance $x$. Importantly, unlike exact duplicates, variants may differ substantially from $x$ in vocabulary, phrasing, or narrative structure, while remaining equivalent in required knowledge, logical dependencies, and reasoning trajectory.

The goal of the VCD task is thus to identify, within a model's test set, which instances $x$ have been subject to contamination by variants present in training.

## 3.2 Benchmark Construction

Variant contamination commonly arises in the context of data augmentation. To systematically evaluate the extent of variant contamination in large language models (LLMs), we construct a dedicated benchmark dataset. The construction leverages mainstream data augmentation techniques Shorten & Khoshgoftaar (2019); Shorten et al. (2021); Maharana et al. (2022) and incorporates two widely used benchmarks: Omni-Math Gao et al. (2024) and SuperGPQA Du et al. (2025). As illustrated in Figure 2, we employ GPT-4o Hurst et al. (2024) as the generation engine to produce semantically equivalent variants from the original problem.

**Initial Question and Solution Verification** In the initial stage, we first verify the original problem–solution pair $(x, y)$. If the problem already comes with a standardized solution, it directly proceeds to the next step; otherwise (e.g., in the SuperGPQA dataset), GPT-4o Hurst et al. (2024) is employed to generate gold-standard answers. This ensures that each problem is paired with a reference solution, providing the foundation for subsequent variant generation. Formally, given the training set:

$$D = \{(x_i, y_i)\}_{i=1}^{N}, \tag{2}$$

we take $(x_i, y_i)$ as input in preparation for generating corresponding variants.

**Variant Generation** In this stage, we adopt mainstream data augmentation techniques Shorten & Khoshgoftaar (2019); Shorten et al. (2021); Maharana et al. (2022) to generate a set of semantically equivalent variants $(x_v, y_v)$ for each original problem (see Table 1 and Figure 7). Specifically, we define a transformation set:

$$T = \{T_{\text{ent}}, T_{\text{scn}}, T_{\text{num}}, T_{\text{nar}}\}, \tag{3}$$

covering four categories: entity substitution, scenario conversion, numerical rewriting, and narrative restructuring. Through these surface-level transformations, we construct the variant set:

$$V(x_i) = \{v_i^{(1)}, \ldots, v_i^{(m)}\}, \quad \text{where } f(v_i^{(j)}) = f(x_i). \tag{4}$$

To guarantee semantic equivalence and correctness, rejection sampling is applied during generation (see Figure 4), with GPT-4o providing candidate variant answers.

**Variant Filter** Finally, GPT-4o is employed as a filter to conduct quality control over the generated variants Liu et al. (2025). The filtering procedure consists of two steps: first, checking whether the information remains unchanged; second, performing a quality evaluation of the solution. Only when both conditions are satisfied is the variant pair $(x_v, y_v)$ accepted. Ultimately, these high-quality variant samples are injected into the training set to simulate test contamination, enabling systematic evaluation of whether existing detection methods can accurately identify variant-contaminated test instances.

## 4 Method

This paper proposes a method named DVD (**D**etection via **V**ariance of generation **D**istribution) grounded in modeling the distribution of model outputs. The core idea is to generate multiple responses under a fixed prompt using temperature sampling, thereby capturing fluctuations in low-probability regions of the model's output distribution. These fluctuations serve as key signals for

Table 1: Variant generation strategies used to simulate contamination.

| Method | Description |
|---|---|
| Entity substitution | Replace referents, variable names, and object categories while maintaining consistency in type and context. |
| Scenario transformation | Alter the background setting and narrative context, while preserving logical dependencies and constraint structures. |
| Numerical rewriting | Resample parameters under solvability constraints and update derivations and intermediate values for consistency. |
| Narrative structure transformation | Rearrange syntax or rewrite step-by-step analysis into a paragraph-style narrative while preserving semantic meaning. |

detecting contamination. More specifically, when a test sample appears in the training set, the model may operate in two distinct generative states. The first is memory adherence, where generation is guided by memorized templates or fragments internalized during training. The second is perturbation drift, where generation is primarily driven by stochastic perturbations introduced by temperature sampling, leading to free-form exploratory outputs. Memory adherence reflects the model's reliance on training-based recall, while perturbation drift captures the natural randomness of unconstrained generation. If a test sample is contaminated, the model alternates between these two states, producing substantial variability in the conditional likelihoods of low-probability tokens. In contrast, for uncontaminated samples, the absence of reliable memory templates constrains the model to remain in a perturbation drift state, where tail-token probabilities mainly reflect inherent noise and thus exhibit only minor fluctuations. Based on this observation, we design the variance of synthetic difficulty as the contamination detection criterion.

## 4.1 Temperature Sampling

For each test sample $x_i$, we apply temperature sampling under a fixed prompt $p$ to generate $N$ candidate responses $\{a_i^{(1)}, a_i^{(2)}, \ldots, a_i^{(N)}\}$. Each response is concatenated with the prompt to form the complete input:

$$s_i^{(k)} = (p, a_i^{(k)}). \tag{5}$$

Temperature sampling introduces controlled stochastic perturbations, enabling the collection of diverse outputs for the same test sample. In uncontaminated cases, generation consistently remains in a perturbation drift state, and temperature perturbations do not substantially alter the statistics of low-probability tokens. In contaminated cases, however, generation alternates between memory adherence and perturbation drift. Here, temperature perturbations amplify the disparity between template-based and non-template-based responses, causing tail tokens to exhibit more pronounced fluctuations.

## 4.2 Synthetic Difficulty Modeling

To quantify such fluctuations, we define the notion of **synthetic difficulty**. For each generated sequence $s_i^{(k)}$, we select the $k$ least probable tokens in the response, compute the sum of their log-likelihoods, and normalize by sequence length $T_i^{(k)}$:

$$D_i^{(k)} = \frac{1}{T_i^{(k)}} \sum_{j=1}^{k} \log P_\theta\left(t^{(j)} \mid s_i^{(k)}\right). \tag{6}$$

This statistic captures local uncertainty in the tail region of the distribution. Unlike global perplexity, tail-token probabilities are more sensitive to the presence of training-set memorization. If a test sample is contaminated, $D_i^{(k)}$ varies markedly across generations due to the alternation between memory adherence and perturbation drift. If uncontaminated, tail probabilities primarily reflect noise, yielding relatively stable values of $D_i^{(k)}$ across multiple generations.

Given the synthetic difficulty set $\{D_i^{(1)}, D_i^{(2)}, \ldots, D_i^{(N)}\}$, we define the DVD indicator as their sample variance:

$$\text{DVD}_i = \frac{1}{N} \sum_{k=1}^{N} \left( D_i^{(k)} - \overline{D}_i \right)^2, \quad \overline{D}_i = \frac{1}{N} \sum_{j=1}^{N} D_i^{(j)}. \tag{7}$$

This indicator effectively characterizes the fluctuation of synthetic difficulty. According to the variance decomposition principle for mixture distributions, if a test sample is contaminated, the distribution of synthetic difficulty can be regarded as a mixture of memory states and drift states, which differ in expectation, thereby inflating the overall variance. If uncontaminated, synthetic difficulty arises from a single state, and variance remains low.

More specifically, contaminated samples can be modeled as a mixture of two latent generation states: the memory-adhering state ($Z = M$) dominated by training memorization, and the unconstrained perturbation-drift state ($Z = U$). Let $\pi_M = \text{Pr}(Z = M)$, $\pi_U = \text{Pr}(Z = U)$, with $\pi_M + \pi_U = 1$. Then,

$$\mu = \pi_M \mu_M + \pi_U \mu_U, \tag{8}$$

$$\text{Var}(X) = \pi_M \left( \sigma_M^2 + (\mu_M - \mu)^2 \right) + \pi_U \left( \sigma_U^2 + (\mu_U - \mu)^2 \right). \tag{9}$$

Here, $\mu_M$ and $\mu_U$ denote the expectations under the memory and drift states, respectively. Since the memory state relies on templates encountered during training, its synthetic difficulty is generally lower than that of the drift state, i.e., $\mu_M > \mu_U$ empirically. By the decomposition of within-group and between-group variance, if the two states differ substantially in expectation, the overall variance of the mixture will necessarily exceed that of a single distribution. This theoretical grounding demonstrates the effectiveness of our method in distinguishing contaminated from uncontaminated samples.

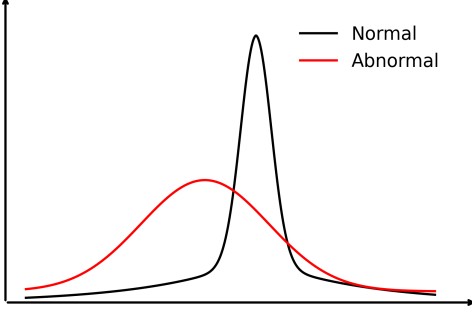

Figure 3: Comparison of Synthetic Difficulty Variance in Normal vs. Abnormal (Contaminated) Samples

Please determine whether the following variant statement is logically correct and free of obvious errors:

**Variant statement:** [Variant statement content]

Next, please assess whether the variant statement is logically equivalent to the original statement and whether it shares the same core solution space:

**Original statement:** [Original statement content]

**Variant statement:** [Variant statement content]

Finally, respond with either "yes" or "no." If any part of the judgment is "no," the final output should be "no."

Figure 4: The prompt we use to do rejection sampling from GPT-4o

## 5 EXPERIMENTS

In this section, we simulate a variant contamination environment based on the constructed variant dataset 3.2 and conduct comparative experiments between the proposed method and several baseline approaches under this setting. The detailed experimental setup is provided in 5.1, and the experimental results are presented in 5.2.

### 5.1 EXPERIMENTAL SETUP

**Model Selection:** To comprehensively evaluate the robustness of the proposed variant contamination detection method, we compare models in the dimension of different parameter scales (Qwen2.5-1.5B-InstructTeam (2024) vs Qwen2.5-3B-InstructTeam (2024) vs Qwen2.5-7B-InstructTeam (2024)) as well as of different architectures (Qwen2.5-7B-Instruct vs Llama3.1-8B-InstructDubey et al. (2024)).

Table 2: Performance comparison of different detection methods on the Omni-MATH and SuperGPQA datasets

| Method | Omni-MATH | | | | SuperGPQA | | | |
|---|---|---|---|---|---|---|---|---|
| | Qwen1.5B | Qwen3B | Qwen7B | Llama8B | Qwen1.5B | Qwen3B | Qwen7B | Llama8B |
| Min-K% | 0.635 | 0.654 | 0.533 | 0.560 | 0.435 | 0.431 | 0.447 | 0.495 |
| Perplexity | 0.597 | 0.620 | 0.582 | 0.591 | 0.464 | 0.462 | 0.473 | 0.498 |
| CDD | 0.509 | 0.495 | 0.550 | 0.495 | 0.585 | 0.586 | 0.663 | 0.686 |
| Embedding-similarity | 0.521 | 0.554 | 0.569 | 0.557 | 0.613 | 0.593 | 0.649 | 0.683 |
| DVD (Ours) | 0.649 | 0.677 | 0.617 | 0.662 | 0.668 | 0.709 | 0.697 | 0.710 |

*Note: The full names of the columns are as follows. Qwen1.5B: Qwen2.5-1.5B-Instruct, Qwen3B: Qwen2.5-3B-Instruct, Qwen7B: Qwen2.5-7B-Instruct, Llama8B: Llama3.1-8B-Instruct. The values in the table represent performance metrics (e.g., AUC), where higher values are better. Background colors are used to highlight the best-performing (pink) and second-best (orange) values in each column.*

**Fine-tuning Details:** To simulate the scenario of variant contamination, we fine-tune the aforementioned models on the constructed variant contamination dataset. All models are fine-tuned using the LoRA approach for parameter-efficient adaptation, with training conducted on a single NVIDIA A800 GPU. The specific configuration is as follows: the LoRA rank is set to 8, the optimizer is Adam, the total number of training epochs is 10, and the initial learning rate is 1e-4. A cosine learning rate scheduler with a warm-up ratio of 0.1 is employed. The batch size per GPU is 2, with a gradient accumulation step of 1. Training is conducted with bfloat16 precision.

**Baseline Methods:** To assess the effectiveness of the proposed method, we compare it against the following baselines: 1) *Embedding Similarity Dong et al. (2024)*: Computing the similarity between answers using embeddings generated by the base model; 2) *Perplexity Li et al. (2023)*: Calculating the perplexity of the original answer under the given prompt; 3) *Min-k% Probability Shi et al. (2023)*: Calculating the minimum k% probability of the original answer under the given prompt; 4) *CDD Dong et al. (2024)*: Measuring the sharpness of the output distribution based on edit distance. All hyperparameters of the baseline methods are consistent with those reported in their original papers. The hyperparameters specific to our method were set as follows: the number of minimum-probability tokens $k$ was fixed at 20, and the number of samples $N$ was set to 50.

## 5.2 EXPERIMENTAL RESULTS

We construct variant data on two datasets from different domains, *Omni-Math* and *SuperGPQA*, to simulate the scenario of variant contamination. The specific construction method is detailed in 3.2. The experimental results are presented in Table 2, where our proposed method consistently and significantly outperforms all baseline approaches across various evaluation metrics, while also demonstrating strong cross-domain generalization capability.

**Outperforms Log-Probability Approaches** The *Min-k% Probability* and *Perplexity* methods can partially identify variant contamination on the Omni-Math dataset. However, their performance degrades substantially on the SuperGPQA dataset, with AUC values dropping below 0.5 and sometimes even showing opposite trends compared to Omni-Math. This indicates that such methods lack cross-domain robustness, as they rely on the model's log-probabilities of standard answers—a mechanism that proves unstable under variant contamination. In contrast, our method avoids this reliance and thus maintains stable performance across domains.

**Surpasses Shallow Distributional Measures** The *CDD* method relies solely on text edit distance to measure distributional differences, which is overly superficial. On the inherently diverse Omni-Math dataset, its performance is close to or even worse than random guessing (e.g., AUC = 0.495 on Llama3.1-8B-Instruct). On SuperGPQA, it shows some improvement (0.663–0.686), but remains clearly inferior. The newly introduced *Embedding-similarity* baseline leverages vector similarity to capture semantic-level distributional shifts. It outperforms CDD (0.521–0.569 on Omni-Math, 0.613–0.683 on SuperGPQA) and becomes the strongest baseline besides ours. Nevertheless, com-

pared to such shallow or surface-level distributional modeling, our method consistently achieves more stable and universal detection performance.

**Maintains Robustness Across Model Scales**  As shown in Table 2, we further examine the performance of different model sizes within the Qwen2.5 family (1.5B, 3B, and 7B). Results demonstrate that our method consistently outperforms all baselines with minimal fluctuations, highlighting its robustness. For example, on the SuperGPQA dataset, our method achieves AUC scores of 0.668, 0.709, and 0.697 on Qwen2.5-1.5B, 3B, and 7B, respectively—all substantially higher than baseline methods. In contrast, Min-k and Perplexity remain in the 0.43–0.47 range, CDD improves slightly but remains insufficient, while Embedding-similarity performs relatively stronger but still falls significantly behind our method.

**Overcomes the Limitations of Small-Scale Models**  We also observe an interesting phenomenon: the CDD method consistently underperforms when model parameter scales are small, while Embedding-similarity is relatively more stable but still insufficient on Omni-Math. This confirms our argument that relying solely on surface-level measures (whether edit distance or vector similarity) is inadequate for capturing deeper distributional shifts in contaminated models. In contrast, our method, inspired by perplexity-based modeling, effectively characterizes output distributional features and achieves stable performance across both different scales and architectures (Qwen and Llama). For instance, on Omni-Math, CDD yields an AUC of only 0.495 on Llama3.1-8B-Instruct, while Embedding-similarity slightly improves it to 0.557, but our method significantly increases it to 0.662.

**Stable and Effective Cross-Domain Detection by Our Method**  In summary, our method not only demonstrates strong generalization across tasks (mathematical reasoning and general reasoning) but also maintains robust advantages across model scales (1.5B to 7B) and architectures (Qwen and Llama). This highlights its capability to achieve stable and effective contamination detection in multi-dimensional scenarios, thereby validating its robustness and practical value. Quantitative comparisons further reveal that, depending on dataset–model combinations, our method achieves AUC improvements of more than 0.2 over the best baseline in some cases, while always maintaining positive gains in all settings. These results strongly support the effectiveness and universality of the proposed approach in variant contamination detection.

## 6 CONCLUSION

This paper systematically revealed the overlooked problem of variant contamination in large language models and proposed DVD as a principled solution. By modeling the fluctuation of synthetic difficulty across multiple generations, our method effectively distinguishes contaminated from uncontaminated samples, overcoming the limitations of perplexity- and similarity-based approaches.

Experiments on Omni-MATH and SuperGPQA demonstrate that DVD consistently outperforms existing baselines across domains, model scales, and architectures, while exhibiting strong robustness to hyperparameter choices. These findings establish DVD as a reliable tool for mitigating contamination risks and ensuring fairer, more trustworthy evaluation of LLM capabilities.

## 7 ETHICS STATEMENT

This work investigates the problem of variant contamination in large language models (LLMs) and proposes a novel detection framework (DVD). All experiments are conducted on publicly available benchmarks (Omni-MATH and SuperGPQA) and synthetic variant data generated through controlled transformations with automated verification; no human subjects, private data, or personally identifiable information were involved. The proposed approach aims to improve the reliability and fairness of LLM evaluation by mitigating risks of data leakage and inflated benchmark performance. While our method enhances the trustworthiness of evaluation, it does not eliminate all forms of bias or contamination inherent in training data. We affirm that our research complies with the ICLR Code of Ethics and does not pose foreseeable harm to individuals or groups.

## 8 REPRODUCIBILITY STATEMENT

To ensure reproducibility and transparency of our results, we have submitted all necessary code and evaluation scripts as supplementary materials, together with detailed instructions to reproduce the experiments reported in this paper.

## 9 THE USE OF LARGE LANGUAGE MODELS (LLMS)

LLMs were used to support the writing process of this paper. Specifically, they assisted in grammar correction, wording refinement, and formatting adjustments. In addition, LLM agents were leveraged to facilitate literature search and provide coding suggestions for implementation. The use of AI tools does not affect the originality of the work or the authors' responsibility for its content.

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

# A APPENDIX

## A.1 HYPERPARAMETER SENSITIVITY ANALYSIS

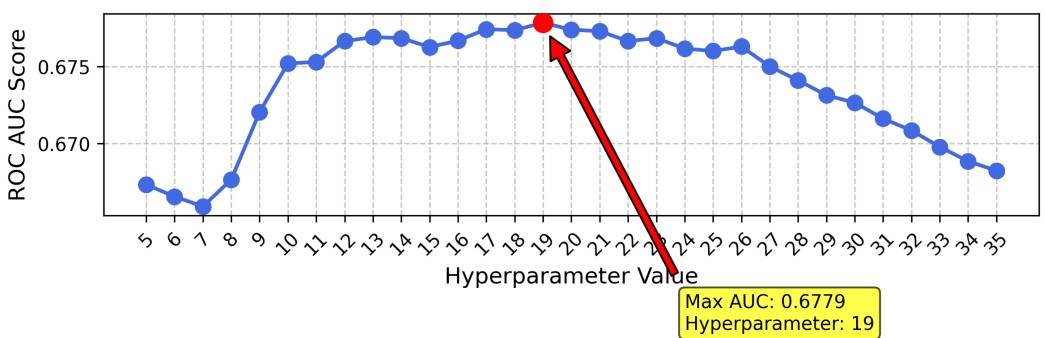

Figure 5: The DVD method demonstrates remarkable robustness across a wide range of hyperparameters.

To evaluate the sensitivity of the proposed DVD method to the key hyperparameter M (i.e., the minimum number of low-probability tokens considered when calculating the synthetic difficulty), we conducted extensive experiments on the Qwen2.5-3B-Instruct model and the Omni-MATH variant dataset. The experimental results (figure5) clearly reveal the robustness characteristics of the method.

**Superior Stability of Performance** The experimental results indicate that the detection performance of the DVD method (measured by AUC) exhibits high stability across an extremely broad range of hyperparameters. As the figure5 show, when M increases from 9 to 28, the AUC values remain within the range of 0.672 to 0.678, with a very small fluctuation range ($< 0.006$). Even when the parameter range is extended from M=5 to M=35, the difference between the global maximum and minimum AUC values is only 0.011 (0.667 to 0.678). This phenomenon of performance flattening across 24 consecutive parameter points indicates that the DVD method is insensitive to the specific value of the hyperparameter M.

**Presence of a Robust Plateau Interval** The performance curve reveals a significant robust plateau interval $M \in [9, 28]$. Within this interval, the AUC values fluctuate slightly around the mean of 0.676, without any noticeable performance peaks or sharp declines. It implies that users do not need to conduct fine-tuning of the hyperparameter, which is time-consuming, but can simply select any value of M within the broad plateau interval to ensure consistent and superior detection performance.

**Comparative Advantage Over Baseline Methods** It is worth noting that even the "stable performance" (AUC $\approx 0.676$) achieved within this broad parameter range is significantly and consistently better than all baseline methods (Min-K% Prob: 0.635, Perplexity: 0.597, CDD: 0.509). This strongly demonstrates that the superiority of the DVD method does not rely on the accidental discovery of a specific lucky parameter, but is an inherent and reproducible characteristic of its intrinsic mechanism.

**Theoretical Implications** This robustness feature is consistent with the core idea of the DVD method. The method detects memory effects by evaluating the variance of "synthetic difficulty" when the model generates answers. The robustness of the hyperparameter $M$ indicates that as long

as a sufficient number ($M \geq 9$) of low-probability tokens are captured to form a representative "difficulty" estimate, its variance is sufficient to effectively distinguish whether the data is contaminated. Over-increasing $M$ ($M > 28$) introduces too many medium-probability tokens, slightly diluting the core signal, leading to a slow decline in performance. However, the performance decay remains very limited even up to $M = 35$, further confirming the stability of the method.

## A.2 CASE STUDY

The three representative cases examined above provide a mechanistic explanation for the macroscopic performance trends observed in Table 2. They demonstrate that the effectiveness of a detection method is not arbitrary but is determined by the intrinsic alignment between its underlying mechanism and the nature of the contamination. The superior performance of our DVD method stems from its unique capacity to probe the model's internal "cognitive state," enabling it to penetrate surface-level textual variations and identify the essential signal of memorization.

| Original Question | Variant Promblem | Transform method | Detection method |
|---|---|---|---|
| 16 students took part in a competition. All problems were multiple choice style. Each problem had four choices. It was said that any two students had at most one answer in common, find the maximum number of problems? | A factory has 20 robots. In each test, robots choose one of 5 task modes (M1–M5).Across all tests, any two robots may share the same mode at most once.What is the maximum number of tests possible? | Entity Substitution: ✔

Scenario conversion: ✔

numerical rewriting: ✔

narrative restructuring: ✔ | Ours: Yes

CDD: No

Min-k: No

Perplexity: No

Embedding Similarity: No |
| Find, with proof, the maximum positive integer $\(k\)$ for which it is possible to color $\(6k\)$ cells of a $\(6 \times 6\)$ grid such that, for any choice of three distinct rows $\(R_{1}, R_{2}, R_{3}\)$ and three distinct columns $\(C_{1}, C_{2}, C_{3}\)$, there exists an uncolored cell $\(c\)$ and integers $\(1 \leq i, j \leq 3\)$ so that $\(c\)$ lies in $\(R_{i}\)$ and $\(C_{j}\)$. | A cinema hall has an $\(8 \times 8\)$ seating arrangement. To maintain safe distancing, the staff may seat at most $\(8k\)$ spectators. Find and prove the maximum positive integer $\(k\)$ such that, for any choice of four rows and four columns, there is always at least one empty seat located at their intersection. | Entity Substitution: ✔

Scenario conversion: ✔

numerical rewriting: ✔

narrative restructuring: ✔ | Ours: Yes

CDD: No

Min-k: No

Perplexity: No

Embedding Similarity: No |
| Given that $a,b,c,d,e$ are real numbers such that\n$a+b+c+d+e=8$ ,\n$a^2+b^2+c^2+d^2+e^2=16$ .\nDetermine the maximum value of $e$ .", | Suppose that five real numbers p, q, r, s, t satisfy p+q+r+s+t=10 and p²+q²+r²+s²+t²=20. Find the maximum value of t. | Entity Substitution: ✔

Scenario conversion: ✘

numerical rewriting: ✔

narrative restructuring: ✘ | Ours: Yes

CDD: Yes

Min-k: No

Perplexity: No

Embedding Similarity: Yes |

Figure 6: Compare the effectiveness of different detection methods on different variants

### A.2.1 DEEPLY TRANSFORMED VARIANTS

Cases 1 and 2 represent deeply transformed variants where all four transformation methods (entity substitution, scenario conversion, etc.) are applied. While the surface narratives are entirely different ("student competition" to "factory robots", "grid coloring" to "cinema seating"), the core mathematical structures are isomorphic.

**CDD** fails because and it is trapped at the surface symbolic level of the text and cannot reach the deep logical equivalence relationships. When the model generates based on these "completely different prompts" at different temperatures, the resulting "inference process text" will inevitably be completely different. For example, the description of the steps for proving the existence of "grid coloring" is almost without overlap in terms of vocabulary and syntax compared to the description of the steps for arranging "cinema seats" with safe distances. The CDD mechanism interprets this as "output inconsistency", and thus it is judged as "unremembered".

**Min-k% and Perplexity** fail as the models' token-level probabilities are sensitive to the unfamiliar surface text, obscuring any signal from the underlying memorized logic.

**Embedding Similarity** fails because standard text embeddings cannot reliably capture the abstract, structural isomorphism of the mathematical problems. It difficult to comprehend that the structure

where "any two student answers can be the same at most on one question" and "any two robot task patterns can be the same at most once" is completely equivalent in combinatorial mathematics.

**The DVD method** is successful by probing the internal cognitive state of the model during generation, rather than analyzing the output text. Despite the surface differences, the model has memorized the core logical template for solving these problem types. When generating answers, it exhibits high confidence at the key reasoning steps and final answer. This results in low and stable "constitutive difficulty" values across samples, leading to a high variance score. Thus, DVD effectively detects contamination by identifying the model's familiarity with the underlying mathematical structure, bypassing surface-level noise.

### A.2.2 SIMPLY TRANSFORMED VARIANTS

Case 3 is a simple variant that involves only entity substitution and numerical rewriting. The mathematical problem (an application of the Cauchy-Schwarz inequality) remains identical, with only variable names and constants changed.

**CDD and Embedding Similarity** succeed in Case 3, due to the high degree of textual and semantic similarity between the original and variant prompts.

**DVD** also succeeds, as the model confidently applies the memorized solution template, resulting in high variance in generation confidence.

**Min-k% / Perplexity** fail, highlighting their fragility. Changes in specific tokens (variables, numbers) are sufficient to alter the probability distributions. Change a, b, c, d, e to p, q, r, s, t, and change 8 and 16 to 10 and 20. These specific token changes are sufficient to significantly alter the distribution of the model's calculation of the probability of the entire sequence. The model has seen $a + b + c + d + e = 8$, but has not seen $p + q + r + s + t = 10$, so it believes that the probability of the latter sequence is slightly lower.

### A.3 PROMPT

You will be provided with a problem in JSON format, with each item separated by a newline. Please generate 4 different variants of the given problem using the following methods:

**1.Entity substitution**: Replace referents, variable names, and object categories while maintaining consistency in type and context.

**2.Scenario transformation**: Alter the background setting and narrative context, preserving logical dependencies and constraint structures.

**3.Numerical rewriting**: Resample parameters under solvability constraints and update derivations and intermediate values for consistency.

**4.Narrative structure transformation**: Rearrange syntax or rewrite step-by-step analysis into a paragraph-style narrative while preserving semantic meaning.

Your response should consist only of newline-delimited JSON format text.

Figure 7: The prompt we use to generate variants fromr asking GPT-4o.

### A.4 STATISTICAL EVIDENCE FOR GENERATION STATES: MEMORY ADHERENCE AND PERTURBATION DRIFT

This section provides the detailed statistical and experimental foundation for introducing the core generation states: **Memory Adherence** and **Perturbation Drift**. These states are not theoretical assumptions but stable, objectively observed modes of behavior resulting from a systematic statistical analysis of the model's generation process on contaminated and uncontaminated samples.

EXPERIMENTAL SETUP

To quantify the model's generation mechanism, we performed multiple repeated samplings for 50 randomly selected samples (including both contaminated and clean examples) at a fixed sampling temperature $\tau$ (e.g., $\tau = 0.8$). The primary statistical quantity analyzed is the distribution of the **log-likelihood of the sum of K-min token** at each generation step. This distribution characterizes the model's propensity to generate tokens with varying degrees of confidence and quality.

BIMODAL STRUCTURE IN CONTAMINATED SAMPLES

For samples subject to variant contamination, the distribution of the sum of K-min token log-likelihood consistently exhibits a **pronounced and repeatable bi-modal structure**. This characteristic structure is direct evidence that the model's generation process, when exposed to contamination, is not governed by a single random mechanism but dynamically switches between two distinct modes.

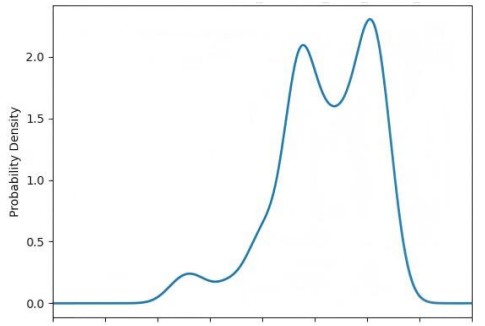 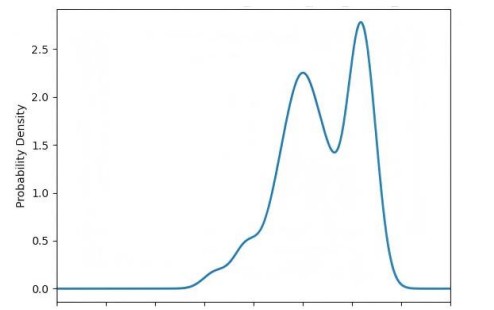

Figure 8: Log-likelihood Distribution of Contaminated Sample (Ex. 1)

Figure 9: Log-likelihood Distribution of Contaminated Sample (Ex. 2)

The analysis of the bi-modal structure reveals the following:

1. **The First Peak (High-Confidence Region):** This peak is consistently located in the **higher** log-likelihood region. It corresponds to generation where the model selects high-probability, high-confidence tokens. This behavior indicates that the model is **adhering to answer fragments, linguistic patterns, or templates** encountered during training. We define this as the **Memory Adherence State**.

2. **The Second Peak (Low-Confidence Region):** This peak resides in the **lower** log-likelihood region, corresponding to the selection of low-probability, high-randomness tokens. This mode suggests the model has **deviated from the memory track** and entered a more explorative, lower-confidence generation space, which we term the **Perturbation Drift State**.

The bi-modal distribution directly proves that the model dynamically alternates between leveraging specific, strong memory structures and engaging in randomized, exploratory sampling on the same contaminated inputs.

UNIMODAL STRUCTURE IN UNCONTAMINATED SAMPLES

In stark contrast, uncontaminated (clean) samples, used as a control, consistently exhibit a **single, smooth, and approximately Gaussian distribution** (Figures A.4).

The unimodal nature confirms that the model is following a **consistent, intrinsic random generation mechanism** without the disruptive influence of strong, pulling memory structures. The absence of a secondary peak supports the hypothesis that state-switching behavior is unique to contaminated data.

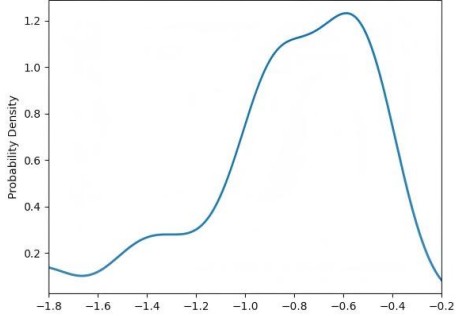

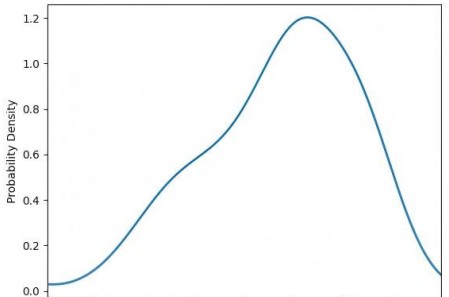

Figure 10: Log-likelihood Distribution of Clean Sample (Ex. 1)

Figure 11: Log-likelihood Distribution of Clean Sample (Ex. 2)

