# OpenReview forum: "Detecting Variant Contamination in LLMs via Variance of Generation Distribution"
_ICLR.cc/2026/Conference — ICLR 2026 Conference Withdrawn Submission_

### Official Review · Reviewer_3LXG · 2025-10-25

**Soundness:** 3
**Presentation:** 3
**Contribution:** 2
**Rating:** 4
**Confidence:** 3

**Summary:**

This paper proposes a sampling-based method to detect variant contamination through the distribution variance of the likelihood of generated tokens. It also provides benchmarks for variant contamination detection.

**Strengths:**

1. The writing of this paper is easy to follow.

2. The construction of the benchmark dataset is helpful for evaluating the proposed task.

**Weaknesses:**

1. My main concern is about the introduction of memory adherence and perturbation drift states. Is there any evidence that the model experiences these two states when conducting temperature scaling? Any flag for differentiating these two states? The authors should demonstrate the existence of the states before they design the method. Moreover, I think the introduction of the states should be at the beginning (e.g., introduction section), but in this paper, the detailed explanation is in section 4.

2. I am wondering if there is any evidence that the training corpus of current LLMs contains variant contamination. It is important to validate the motivation with references.

3. The authors only conduct experiments on the self-created benchmark. Although the authors claim that this paper is about variant contamination. More experiments on common data contamination datasets such as WIKIMIA.

**Questions:**

Please refer to the weakness section.

---

> ### Author Response · Authors · 2025-12-04
>
> > W1:My main concern is about the introduction of memory adherence and perturbation drift states. Is there any evidence that the model experiences these two states when conducting temperature scaling? Any flag for differentiating these two states? The authors should demonstrate the existence of the states before they design the method. Moreover, I think the introduction of the states should be at the beginning (e.g., introduction section), but in this paper, the detailed explanation is in section 4.
>
> Regarding the rationale and evidence for the two generated states you raised: 'memory adherence' and 'perturbation drift', we have moved this content to the appendix of the rebuttal revision version.
>
> > W2: I am wondering if there is any evidence that the training corpus of current LLMs contains variant contamination. It is important to validate the motivation with references.
>
> Thank you for your question, Reviewer. Regarding the evidence for whether **"variant contamination exists in current LLM training corpora,"** multiple studies have explicitly indicated that existing decontamination methods based on n-gram or character matching cannot identify semantic, structural, or variant-level contamination, and that this type of contamination objectively exists in real training data and has caused severe impacts.
>
> * Firstly, Yang et al. demonstrated that simple test data variants (such as synonym paraphrasing or translation) can easily bypass existing decontamination measures. Specifically, the authors found an 8–18% overlap of the HumanEval benchmark data in pretraining sets like RedPajama-Data-1T and StarCoder-Data. More interestingly, they also discovered such contamination in synthetic data generated by GPT-3.5/4, suggesting a potential risk of unintentional contamination.
> * Secondly, Mehta et al. further pointed out that existing detection methods can only identify token-level overlap but fail to recognize semantic-level contamination. In controlled experiments on MMLU, GSM8K, and HumanEval, semantic variants could easily bypass all existing detectors with a detection rate of $F1=0.17-0.49$, yet these samples still substantially boost model scores on the corresponding benchmarks. This demonstrates that semantic variant contamination is empirically confirmed to exist and genuinely threatens evaluation reliability.
>
> Building upon this evidence, our work is the first to systematically address this type of contamination, which is difficult for existing methods to capture. We explicitly propose a unified definition of **"variant contamination,"** encompassing the broad categories of paraphrasing, translation, structural transformation, and style transfer, and we demonstrate its impact through large-scale experiments. In other words, we are not hypothesizing this risk arbitrarily, but rather, based on existing literature reporting numerous real contamination cases, we are the first to unify these scattered phenomena conceptually as "variant contamination" and provide a systematic method for its detection and quantification.
>
> Therefore, multiple independent studies support the existence of variant contamination in current LLM training corpora, and our work builds on this evidence to further organize the problem, unify the concept, and propose a more comprehensive detection framework.
>
> References
>
> [1] Yang S, Chiang W L, Zheng L, et al. Rethinking benchmark and contamination for language models with rephrased samples[J]. arXiv preprint arXiv:2311.04850, 2023.
>
> [2] Mehta S. Beyond Surface-Level Similarity: Hierarchical Contamination Detection for Synthetic Training Data in Foundation Models[J]. arXiv preprint arXiv:2511.17602, 2025.

---

> > ### Author Response · Authors · 2025-12-04
> >
> > > W3: The authors only conduct experiments on the self-created benchmark. Although the authors claim that this paper is about variant contamination. More experiments on common data contamination datasets such as WIKIMIA.
> >
> > Thank you for the reviewer's suggestion regarding the use of existing contamination detection datasets like WIKIMIA. We fully understand the reviewer's motivation to see validation on a broader benchmark. However, WIKIMIA and current mainstream data contamination detection datasets fundamentally differ from the **"variant contamination"** we study in terms of task assumptions and sample format, making them unsuitable for validating this work.
> >
> > Specifically, datasets like WIKIMIA primarily target **exact-match contamination** or near-repetition at the verbatim level. Their detection methods rely on superficial features such as n-gram overlap, embedding similarity, or fuzzy matching. In contrast, the core nature of variant contamination is that the samples appearing in the training set are equivalent to the test samples in terms of semantics, logical dependencies, and reasoning structure, but their superficial form has been significantly altered through transformations like entity replacement, narrative rewriting, or structural adjustments. This type of variant systematically breaks all the superficial similarity cues that WIKIMIA relies on (n-gram, embedding distance, text overlap, etc.), making it impossible to be identified by its "verbatim or near-verbatim repetition detection" framework. In other words, WIKIMIA was designed to determine if a piece of text had similar phrasing appear in the training set, not to determine if the model had previously encountered a **"semantically and structurally equivalent but rewritten variant."** Therefore, WIKIMIA's detection target does not align with the variant contamination scenario we define and cannot provide an effective evaluation for this task.

---

### Official Review · Reviewer_on2q · 2025-10-29

**Soundness:** 1
**Presentation:** 3
**Contribution:** 2
**Rating:** 2
**Confidence:** 5

**Summary:**

In this work, the authors introduce the problem of "variant contamination", which is a data contamination problem where contaminated samples are semantically equivalent but lexically diverse variants. The authors further propose the Detection via Variance of generation Distribution (DVD) to mitigate this new problem.

**Strengths:**

- The idea of using the model's fluctuation between "memory adherence" and "perturbation drift" is interesting.
- The paper has a nice and steady flow, making it easy to understand the overall idea.

**Weaknesses:**

### 1. Problem Definition
- My major concern is with the problem of "variant contamination". I am not fully convinced that variant contamination is a problem and that it needs to be mitigated. First of all, it is ambiguous to draw a line between two sample variants to say which one is contaminated and which differs significantly from the original question. For example, in the case study in Appendix A.2, the first example shows a pair of problem samples. Honestly, I'd say these two questions are pretty different, even if the core reasoning trace required to solve the problem might be similar. In fact, this type of question variation is exactly the way humans are tested in math classes. Students are taught the core reasoning process, and they are tested on similar question types with different settings and numbers. We don't call that "cheating" -- that's how we "learn".
- That said, for this problem formulation to make sense, I strongly think there needs to be a concrete way to control the level of semantic variation in generating samples. Could the authors provide at least a high-level way to quantify and control the level of semantic variation?

### 2. Fine-tuning Details
The authors simulate the scenario of variant contamination by fine-tuning the models themselves. They use 10 epochs, which, I believe, is an excessively large number of iterations to simulate the real-world contamination environments. Usually, an LLM is trained on a single sample only once or twice. I think 10 epochs will leave a very strong fingerprint on the model and make the task way too easy. Please provide experiments on a setting where the model is trained on the data for 1 epoch.

### 3. Verification of the Core Assumption in Method Design
While the idea of utilizing the model's fluctuation between "memory adherence" and "perturbation drift" is interesting, whether that is actually the case has not been verified. Please provide a comparison between two plots (i.e., with and without variant contamination), where each plot shows the token log likelihood on the y axis and the token index on the x axis. If the authors' assumption is true, there will be occasional basins and plateaus in the log likelihood trend for variant-contaminated samples.

### 4. Model Scales
The authors claim that DVD "maintains robustness across model scales" by demonstrating results on 1.5B, 3B, and 7B. While three variants are usually enough, I suggest trying out 32B if resources permit. In my opinion, an LLM smaller than 7B is weak for reasoning tasks and may not be a good specimen for contamination detection.


### 5. Minor points
- Figure 1 lacks a bit of detail to help understand the content. Please include a caption that explains the figure.

**Questions:**

See Weaknesses

---

> ### Author Response · Authors · 2025-12-04
>
> > W1:Problem Definition
> >
> > - My major concern is with the problem of "variant contamination". I am not fully convinced that variant contamination is a problem and that it needs to be mitigated. First of all, it is ambiguous to draw a line between two sample variants to say which one is contaminated and which differs significantly from the original question. For example, in the case study in Appendix A.2, the first example shows a pair of problem samples. Honestly, I'd say these two questions are pretty different, even if the core reasoning trace required to solve the problem might be similar. In fact, this type of question variation is exactly the way humans are tested in math classes. Students are taught the core reasoning process, and they are tested on similar question types with different settings and numbers. We don't call that "cheating" -- that's how we "learn".
> > - That said, for this problem formulation to make sense, I strongly think there needs to be a concrete way to control the level of semantic variation in generating samples. Could the authors provide at least a high-level way to quantify and control the level of semantic variation?
>
> Thank you very much for your deep consideration and question regarding the concept of "**variant contamination**." We fully understand this concern.
>
> It is true that in human learning, changing numerical values, scenarios, or expressions is often used to test mastery of the same underlying reasoning pattern, and such a "superficial transformation" would not be considered cheating.
>
> However, in the current context of Large Language Model (LLM) research, the community typically regards **"the ability to transcend superficial forms and truly execute cross-semantic generalization"** as the core metric for evaluating a model's generalization capability. For example, several recent works use "whether a model can handle logically equivalent but differently expressed inputs" as the specific criterion for "generalization beyond pattern matching." Our definition in this paper strictly adheres to this widely accepted boundary: if a model has encountered a variant that is structurally equivalent in terms of reasoning to a test sample during the training phase, that sample can no longer be considered a test of **"true generalization ability"** but should be classified as a form of **contamination** in the evaluation.
>
> Furthermore, we completely agree with the reviewer's analogy—simple superficial changes do not constitute "cheating" in the sense of human learning. However, the analogy differs when applied to LLM training: if original problems from benchmarks like GSM8K or AIME are lightly rewritten (e.g., via numerical or entity substitution) and added directly to the training set, the model's excellent performance on corresponding test problems might stem primarily from memory recall of patterns, rather than possessing cross-expression reasoning ability. From an evaluation perspective, this scenario is closer to **data leakage** rather than a normal "learning process." Thus, we include such scenarios within the scope of "variant contamination" discussion.
>
> Regarding the reviewer's suggestion on **"how to quantify and control the degree of semantic change,"** we recognize its importance. Therefore, we adopted a two-layer constraint method in our benchmark construction:
>
> 1.  Using a **semantic abstraction function $f(\cdot)$** to define equivalence conditions, ensuring all variants maintain consistency in their core reasoning structure.
> 2.  Employing a **two-stage filtering mechanism with GPT-4o** (information consistency check + solution correctness verification) to control the magnitude of semantic drift, ensuring variants are neither excessively similar to the original problem nor deviate too far from the original reasoning space.
>
> In the subsequent version, we can report this quantitative strategy for **"controlling the boundary of semantic change"** in more detail.
>
> We thank the reviewer again for their valuable suggestions and are very willing to further clarify this concept in the final version to prevent readers from confusing "human-learning-style problem variation" with "variant leakage in model training."
>
> ---
>
> References
>
> [1] Lunardi R, Della Mea V, Mizzaro S, et al. On Robustness and Reliability of Benchmark-Based Evaluation of LLMs[J]. arXiv preprint arXiv:2509.04013, 2025.
> [2] Sclar M, Choi Y, Tsvetkov Y, et al. Quantifying Language Models' Sensitivity to Spurious Features in Prompt Design or: How I learned to start worrying about prompt formatting[J]. arXiv preprint arXiv:2310.11324, 2023.
> [3] Zhao Y, Yan L, Sun W, et al. Improving the robustness of large language models via consistency alignment[J]. arXiv preprint arXiv:2403.14221, 2024.

---

> > ### Author Response · Authors · 2025-12-04
> >
> > > W2: 2. Fine-tuning Details
> > >
> > > The authors simulate the scenario of variant contamination by fine-tuning the models themselves. They use 10 epochs, which, I believe, is an excessively large number of iterations to simulate the real-world contamination environments. Usually, an LLM is trained on a single sample only once or twice. I think 10 epochs will leave a very strong fingerprint on the model and make the task way too easy. Please provide experiments on a setting where the model is trained on the data for 1 epoch.
> >
> > Thank you for the valuable feedback, Reviewer. We completely understand your concerns regarding the setting of fine-tuning epochs on potentially polluted data. While our paper presents the results of 10 epochs of training, we conducted experiments across epochs 1 through 10 to more comprehensively validate the model's performance under different contamination levels. The specific data is detailed in our response to **Reviewer-VfGj**, under **W4**.
> >
> > > W3: 3. Verification of the Core Assumption in Method Design
> > >
> > > While the idea of utilizing the model's fluctuation between "memory adherence" and "perturbation drift" is interesting, whether that is actually the case has not been verified. Please provide a comparison between two plots (i.e., with and without variant contamination), where each plot shows the token log likelihood on the y axis and the token index on the x axis. If the authors' assumption is true, there will be occasional basins and plateaus in the log likelihood trend for variant-contaminated samples.
> >
> > Regarding the rationale and evidence for the two generated states you raised: 'memory adherence' and 'perturbation drift', we have moved this content to the appendix of the rebuttal revision version.
> >
> > > W4: 4. Model Scales
> > >
> > > The authors claim that DVD "maintains robustness across model scales" by demonstrating results on 1.5B, 3B, and 7B. While three variants are usually enough, I suggest trying out 32B if resources permit. In my opinion, an LLM smaller than 7B is weak for reasoning tasks and may not be a good specimen for contamination detection.
> >
> > Thank you for your feedback. We have added the corresponding experiments, and the specific data is detailed in our response to **Reviewer-VfGj**, under **We1** and **W3**.
> >
> > > W5: 5. Minor points
> > >
> > > - Figure 1 lacks a bit of detail to help understand the content. Please include a caption that explains the figure.
> >
> > Thank you for your feedback. To help you and other readers better understand our method, we will redraw Figure 1 and include the updated version in the subsequent revision.

---

### Official Review · Reviewer_GMqz · 2025-10-30

**Soundness:** 3
**Presentation:** 2
**Contribution:** 2
**Rating:** 4
**Confidence:** 3

**Summary:**

This paper studies variant contamination, models encounter paraphrased or structurally altered versions of training data that still influence evaluation. The authors prompt LLMs to generate two benchmarks (Omni-MATH and SuperGPQA) by modifying existing datasets to include such variants. They also propose DVD (Detection via Variance of Generation Distribution), which measures the variance of synthetic difficulty across multiple generations under different temperatures. The key claim is that contaminated data cause the model to alternate between memory recall and exploration (drift) states, producing unusually high variance compared with uncontaminated cases. The method is evaluated on the two author-generated benchmarks and compared against perplexity, Min-k%, CDD, and embedding-similarity baselines.

**Strengths:**

* The paper draws attention to the problem of variant contamination and provides two datasets, which could be valuable resources for the community.
* The proposed DVD method is conceptually simple. It only requires repeated sampling under temperature perturbations, making it easy to reproduce and extend.

**Weaknesses:**

* The paper assumes that high generation variance reflects alternation between memory and reasoning states, but provides no behavioral or interpretive evidence that such alternation actually exists. The observed variance difference could arise from other factors, eg overfitting noise or instability in fine-tuned logits.

* The models are fine-tuned for **10 epochs** with lr=1e-4 on contaminated data, simulating heavy retraining rather than realistic light contamination. Such strong exposure likely alters the model’s generation dynamics and confounds the claimed relationship between variance and contamination.

* The motivation of the task rests on an implicit assumption: LLMs lack strong generalization ability on paraphrases -> LLMs will not produce confident or consistent outputs for paraphrased inputs. -> This makes previous contamination detectors (perplexity- or similarity-based) cannot distinguish contaminated from clean variants. However, the authors provide neither empirical evidence nor theoretical justification for this assumption. Without controlled contrasts between paraphrased generalization and memorized variants, the distinctiveness of DVD remains speculative.

**Questions:**

* It would be informative to include a controlled experiment with light contamination, eg training for one epoch with a lr=1e-5, to see whether  DVD still distinguishes contamination from clean data.

* It should include some statistical significance tests for table 2.

* It would strengthen the evaluation to include additional contamination detectors, such as TS-Guessing, exposure-based memorization metrics, or membership inference probes.

---

> ### Author Response · Authors · 2025-12-04
>
> > - W1:The paper assumes that high generation variance reflects alternation between memory and reasoning states, but provides no behavioral or interpretive evidence that such alternation actually exists. The observed variance difference could arise from other factors, eg overfitting noise or instability in fine-tuned logits.
>
> Regarding the rationale and evidence for the two generated states you raised: 'memory adherence' and 'perturbation drift', we have moved this content to the appendix of the rebuttal revision version.
>
> > W2:The models are fine-tuned for 10 epochs with lr=1e-4 on contaminated data, simulating heavy retraining rather than realistic light contamination. Such strong exposure likely alters the model’s generation dynamics and confounds the claimed relationship between variance and contamination
>
> Thank you for the valuable feedback, Reviewer. We completely understand your concerns regarding the fine-tuning settings on potentially polluted data. While our paper presents the results of 10 epochs of training, we conducted experiments across epochs 1 through 10 to more comprehensively validate the model's performance under different contamination levels. The specific data is detailed in our response to **Reviewer-VfGj**, under **W4**.
>
> > W3: The motivation of the task rests on an implicit assumption: LLMs lack strong generalization ability on paraphrases -> LLMs will not produce confident or consistent outputs for paraphrased inputs. -> This makes previous contamination detectors (perplexity- or similarity-based) cannot distinguish contaminated from clean variants. However, the authors provide neither empirical evidence nor theoretical justification for this assumption. Without controlled contrasts between paraphrased generalization and memorized variants, the distinctiveness of DVD remains speculative.
>
> Thank you for raising this question. Regarding the hypothesis you mentioned, it has indeed been experimentally verified by numerous studies.
>
> For example, Lunardi et al. (2025) systematically generated multiple paraphrases for all questions across six common benchmarks and measured the difference in performance among 34 large language models of varying sizes and capabilities on these rephrased questions. The study found that although the relative ranking of LLMs remained stable between paraphrased inputs, their absolute performance scores significantly varied and decreased. Sclar et al. (2023) also found that model performance varies significantly when large language models are faced with different prompt formats. Zhao et al. (2024) quantitatively defined the phenomenon where large language models produce significantly inconsistent responses due to minor changes in wording.
>
> However, we did not explicitly mention this hypothesis in the current version of our paper. We will supplement this content in the subsequent revision to ensure the context of this assumption is clearer and more explicit.
>
> Thank you again for your valuable feedback!
>
> ------
>
> **References**
>
> [1] Lunardi R, Della Mea V, Mizzaro S, et al. On Robustness and Reliability of Benchmark-Based Evaluation of LLMs[J]. arXiv preprint arXiv:2509.04013, 2025. [2] Sclar M, Choi Y, Tsvetkov Y, et al. Quantifying Language Models' Sensitivity to Spurious Features in Prompt Design or: How I learned to start worrying about prompt formatting[J]. arXiv preprint arXiv:2310.11324, 2023. [3] Zhao Y, Yan L, Sun W, et al. Improving the robustness of large language models via consistency alignment[J]. arXiv preprint arXiv:2403.14221, 2024.

---

> > ### Author Response · Authors · 2025-12-04
> >
> > > Q2: It should include some statistical significance tests for table 2.
> >
> > Thank you for the valuable feedback on our work, Reviewer. Regarding your suggestion to include statistical significance testing in the tables, we have performed a significance analysis on the relevant metrics.
> >
> > By conducting statistical tests on the performance of the CDD model and our method (DVD) across different datasets, we confirmed that the performance difference for our DVD method is statistically significant. The specific data is detailed in our response to **Reviewer-VfGj**, under **W2**.
> >
> > > Q3: It would strengthen the evaluation to include additional contamination detectors, such as TS-Guessing, exposure-based memorization metrics, or membership inference probes.
> >
> > Thank you for the valuable feedback on our work, Reviewer. Due to time constraints, we will consider adding a detailed comparison with these methods in a subsequent version of the paper.

---

### Official Review · Reviewer_VfGj · 2025-11-03

**Soundness:** 1
**Presentation:** 3
**Contribution:** 3
**Rating:** 2
**Confidence:** 4

**Summary:**

The paper proposes using the variance of normalized sum of K lowest probability tokens across multiple decodings of the same prompt as a measure to determine whether an example is contaminated or not.

Using K-min tokens is already an explored aspect in membership inference/contamination detection. However they extend this and present an intuitive explanation behind this measure. The explanation is based on two states that models are assumed to be operating under named "memory adherence" and "perturbation drift".

The experimental setup is LoRA finetuning various models on 10 epochs of modified versions of the Omni-Math and SuperGPQA to see if a method can detect contamination in that finetuned model. They experiment with Qwen and LLama models from 1.5B to 8B. The authors create the variants with a verified step which is an extra contribution of the paper.

They find that DVD(proposed method) achieves higher AUC compared to mink-prob, just using perplexity, cdd or embedding similarity.

**Strengths:**

The selected topic is an important topic. Variant contamination is an important topic to that should be explored in more detail. This could help us understand model capabilities and generalization better.
The way the authors create the variant data for Omni-math and SuperGPQA is quite good, this data can be quite useful in future experiments and can be used by the community.
The authors found a creative metric and presented it with an intuitive formulation.

While the topic is important and the idea is quite good if the effectiveness can be shown in a generalizable way in the current state the paper needs to be improved significantly.

**Weaknesses:**

The experimental setup seems to be rather challenging. Contamination in pre-training and finetuning are already known to have relatively large differences between them. This paper uses a LoRA finetuning which might even be more different making the results hard o generalize. While pre-training might be prohibitively expensive at least a full finetuning should be possible with the small models at hand.

There is no statistical tests or confidence intervals that can support the main claim of improvement above the baseline making claims more challenging to be accepted.

While CDD is a relatively recent method min-K++ could have been used instead of min-K% as a stronger baseline.

The results are presented under 10 epochs of finetuning, it is hard to be convinced without seeing results without at least seeing one epochs and also performance of the models on said tasks after finetuning for 1 and 10 epochs.

The paper has lots of repetitive parts particularly around the intuition between memory adherence and perturbation drift modes but there is no reference or foundation of behind these modes. While it is useful for authors to share intuition they have developed during the experiments towards a paper if they are going to be this central to the narrative a concrete grounding in the literature or empirical work around these modes could be expected.

**Questions:**

Do you expect the results to generalize to full finetuning?
Should we expect similar results when there is only one epoch of finetuning instead of 10?
How much do the models actually get better after 1 epoch of finetuning on modified samples? How much after 10?
Are you going to run any significance tests?
Would including stronger baselines be possible?
Is the temparature value provided in the paper? Am I missing something?

---

> ### Author Response · Authors · 2025-12-04
>
> > W1: The experimental setup seems to be rather challenging. Contamination in pre-training and finetuning are already known to have relatively large differences between them. This paper uses a LoRA finetuning which might even be more different making the results hard o generalize. While pre-training might be prohibitively expensive at least a full finetuning should be possible with the small models at hand.
>
> Thank you for your suggestion regarding full parameter fine-tuning. Based on the reviewer's suggestion, we have added more comparative experiments between full parameter fine-tuning and LoRA fine-tuning to better demonstrate the generalization of our method. We conducted 1 epoch of full parameter fine-tuning on the Omni-Math and SuperGPQA datasets, respectively, and provide the relevant experimental results in the tables below. After 1 epoch of full parameter fine-tuning, the 'dvd' method showed high performance across multiple models:
>
> The table below shows the results of 1 Epoch Full Parameter Fine-Tuning on Omni-Math **(lr=1e-5)**
>
> | **Method** | **Qwen2.5-32B-Instruct** | **Qwen2.5-7B-Instruct** | **Qwen2.5-3B-Instruct** | **Qwen2.5-1.5B-Instruct** | **Llama3.1-8B-Instruct** |
> | ---------- | ------------------------ | ----------------------- | ----------------------- | ------------------------- | ------------------------ |
> | dvd        | 0.666505                 | 0.734142                | 0.746926                | 0.743528                  | 0.550388                 |
> | cdd        | 0.506602                 | 0.511748                | 0.495146                | 0.493592                  | 0.492896                 |
> | min-k      | 0.577799                 | 0.571780                | 0.555955                | 0.538382                  | 0.578803                 |
> | perplexity | 0.555663                 | 0.556731                | 0.549515                | 0.543722                  | 0.531845                 |
> | loss       | 0.634887                 | 0.649450                | 0.633883                | 0.625502                  | 0.663948                 |
> | zlib       | 0.580841                 | 0.582977                | 0.576472                | 0.573712                  | 0.560518                 |
> | mink++     | 0.642751                 | 0.646278                | 0.667346                | 0.665146                  | 0.622848                 |
>
> The table below shows the results of 1 Epoch Full Parameter Fine-Tuning on SuperGPQA
>
> | **Method** | **Qwen2.5-32B-Instruct** | **Qwen2.5-7B-Instruct** | **Qwen2.5-3B-Instruct** | **Qwen2.5-1.5B-Instruct** | **Llama3.1-8B-Instruct** |
> | ---------- | ------------------------ | ----------------------- | ----------------------- | ------------------------- | ------------------------ |
> | dvd        | 0.751296                 | 0.693216                | 0.737632                | 0.673904                  | 0.679072                 |
> | cdd        | 0.600232                 | 0.588392                | 0.517144                | 0.520320                  | 0.559552                 |
> | min-k      | 0.341040                 | 0.413936                | 0.519120                | 0.493664                  | 0.389120                 |
> | perplexity | 0.370112                 | 0.407392                | 0.495232                | 0.477984                  | 0.455344                 |
> | loss       | 0.332928                 | 0.326240                | 0.536736                | 0.456608                  | 0.588448                 |
> | zlib       | 0.376192                 | 0.478560                | 0.479248                | 0.497008                  | 0.522880                 |
> | mink++     | 0.374976                 | 0.528912                | 0.460496                | 0.423968                  | 0.540064                 |

---

> > ### Author Response · Authors · 2025-12-04
> >
> > > W2:There is no statistical tests or confidence intervals that can support the main claim of improvement above the baseline making claims more challenging to be accepted.
> >
> > Thank you to the reviewer for the valuable feedback on our work. Regarding your suggestion to include statistical significance testing in the tables, we have performed a significance analysis on the relevant metrics **(Full Parameter Fine-Tuning  lr=1e-5  epoch=10)**.
> >
> > | **Dataset**    | **Omni-MATH**     |                   |                   |                   |                   | **SuperGPQA**     |                   |                   |                    |                   |
> > | -------------- | ----------------- | ----------------- | ----------------- | ----------------- | ----------------- | ----------------- | ----------------- | ----------------- | ------------------ | ----------------- |
> > |                | Qwen1.5B          | Qwen3B            | Qwen7B            | Qwen32B           | Llama8B           | Qwen1.5B          | Qwen3B            | Qwen7B            | Qwen32B            | Llama8B           |
> > | **CDD**        | 0.509 ± 0.018     | 0.495 ± 0.009     | 0.510 ± 0.010     | 0.502 ± 0.02      | 0.495 ± 0.017     | 0.525 ± 0.019     | 0.509 ± 0.016     | 0.586 ± 0.013     | 0.603 ± 0.020      | 0.557 ± 0.015     |
> > | **DVD (Ours)** | **0.742 ± 0.012** | **0.747 ± 0.012** | **0.732 ± 0.008** | **0.750 ± 0.015** | **0.662 ± 0.010** | **0.678 ± 0.018** | **0.739 ± 0.012** | **0.697 ± 0.009** | **0.750 ± 0.0013** | **0.710 ± 0.009** |
> > | **Conf (%)**   | 99                | 99                | 99                | 99                | 99                | 99                | 99                | 99                | 99                 | 99                |
> >
> > By conducting statistical tests on the performance of the CDD model and our method (DVD) across different datasets, we have confirmed that the difference between the DVD method and the baseline method is statistically significant. This further validates the effectiveness of our approach and enhances the reliability of our research findings. We will include these results in the appendix of the paper.

---

> > > ### Author Response · Authors · 2025-12-04
> > >
> > > > W3:While CDD is a relatively recent method min-K++ could have been used instead of min-K% as a stronger baseline.
> > >
> > > Thank you for the valuable feedback, Reviewer. We have added the Min-K++ baseline to our experiments.
> > >
> > > The tables below show the results of 10 Epochs of LoRA Fine-Tuning on Omni-Math and SuperGPQA, respectively.
> > >
> > > Results of 10 Epoch LoRA Fine-Tuning **lr=1e-5**
> > >
> > > Omni-Math Dataset
> > >
> > > | **name**       | **Qwen2.5-32B-Instruct** | **Qwen2.5-7B-Instruct** | **Qwen2.5-3B-Instruct** | **Qwen2.5-1.5B-Instruct** | **Llama3.1-8B-Instruct** |
> > > | -------------- | ------------------------ | ----------------------- | ----------------------- | ------------------------- | ------------------------ |
> > > | **dvd**        | 0.714595                 | 0.731133                | 0.744887                | 0.771230                  | 0.707411                 |
> > > | **cdd**        | 0.500922                 | 0.583333                | 0.502718                | 0.581278                  | 0.600922                 |
> > > | **min-k**      | 0.535663                 | 0.531068                | 0.505405                | 0.549029                  | 0.615502                 |
> > > | **perplexity** | 0.556828                 | 0.566634                | 0.542880                | 0.572071                  | 0.669320                 |
> > > | **loss**       | 0.563042                 | 0.576314                | 0.556117                | 0.571165                  | 0.648350                 |
> > > | **zlib**       | 0.550388                 | 0.549871                | 0.541683                | 0.549191                  | 0.585178                 |
> > > | **mink++**     | 0.594595                 | 0.636117                | 0.620809                | 0.633204                  | 0.670097                 |
> > >
> > > SuperGPQA Dataset
> > >
> > > | **name**       | **Qwen2.5-32B-Instruct** | **Qwen2.5-7B-Instruct** | **Qwen2.5-3B-Instruct** | **Qwen2.5-1.5B-Instruct** | **Llama3.1-8B-Instruct** |
> > > | -------------- | ------------------------ | ----------------------- | ----------------------- | ------------------------- | ------------------------ |
> > > | **dvd**        | 0.742960                 | 0.739664                | 0.707872                | 0.770432                  | 0.714096                 |
> > > | **cdd**        | 0.502056                 | 0.518416                | 0.504424                | 0.496000                  | 0.586728                 |
> > > | **min-k**      | 0.538656                 | 0.511264                | 0.492240                | 0.501264                  | 0.500976                 |
> > > | **perplexity** | 0.517408                 | 0.512832                | 0.520128                | 0.516704                  | 0.511776                 |
> > > | **loss**       | 0.408912                 | 0.407904                | 0.405872                | 0.403520                  | 0.391072                 |
> > > | **zlib**       | 0.427024                 | 0.428416                | 0.425184                | 0.424432                  | 0.416688                 |
> > > | **mink++**     | 0.441672                 | 0.428400                | 0.460144                | 0.425320                  | 0.449360                 |

---

> ### Author Response · Authors · 2025-12-04
>
> > W4: The results are presented under 10 epochs of finetuning, it is hard to be convinced without seeing results without at least seeing one epochs and also performance of the models on said tasks after finetuning for 1 and 10 epochs.
>
> Thank you for the valuable feedback, Reviewer. We completely understand your concerns regarding the fine-tuning settings on potentially polluted data. While our paper primarily shows the results after 10 epochs of training (lr=1e-5), we conducted experiments across epochs 1 through 10 to comprehensively validate the model's performance under various contamination levels.
>
> The table below presents the experimental results on the Omni-MATH dataset using the Qwen2.5-7B-Instruct model under LoRA fine-tuning. These results cover training epochs from 1 to 10, based on experiments conducted with 10 random seeds. We calculated the Mean AUC for both the DVD and CDD methods for each epoch.
>
> The results show that in comparative experiments closer to a real-world scenario (e.g., at 1 epoch), DVD is still effective at identifying polluted data. Furthermore, DVD demonstrates superior performance on contaminated data compared to the CDD method.
>
> | **Epoch** | **Dataset** | **DVD AUC Mean** | **CDD AUC Mean** |
> | --------- | ----------- | ---------------- | ---------------- |
> | 1         | Omni-MATH   | 0.66975728       | 0.49629455        |
> | 2         | Omni-MATH   | 0.64768285       | 0.51623514             |
> | 3         | Omni-MATH   | 0.67097087       | 0.49847896       |
> | 4         | Omni-MATH   | 0.69517152       | 0.51614887       |
> | 5         | Omni-MATH   | 0.6794466        | 0.57928803       |
> | 6         | Omni-MATH   | 0.71703883       | 0.53898585       |
> | 7         | Omni-MATH   | 0.64130421       | 0.56452423       |
> | 8         | Omni-MATH   | 0.68602589       | 0.55194175       |
> | 9         | Omni-MATH   | 0.69288673       | 0.56147249       |
> | 10        | Omni-MATH   | 0.731327         | 0.58365696       |
>
> > W5: The paper has lots of repetitive parts particularly around the intuition between memory adherence and perturbation drift modes but there is no reference or foundation of behind these modes. While it is useful for authors to share intuition they have developed during the experiments towards a paper if they are going to be this central to the narrative a concrete grounding in the literature or empirical work around these modes could be expected.
>
> Regarding the rationale and evidence for the two generated states you raised: 'memory adherence' and 'perturbation drift', we have moved this content to the appendix of the rebuttal revision version.

---

### Note · Authors · 2026-01-06

I have read and agree with the venue's withdrawal policy on behalf of myself and my co-authors.